# Spatial co-occurrence patterns of sympatric large carnivores in a multi-use African system

**Paolo Strampelli**[1,2,3]\*, **Philipp Henschel**[3], **Charlotte E. Searle**[1,2], **David W. Macdonald**[1], **Amy J. Dickman**[1,2]

**1** Department of Zoology, Wildlife Conservation Research Unit, Recanati Kaplan Centre, University of Oxford, Oxford, United Kingdom, **2** Lion Landscapes, Iringa, Tanzania, **3** Panthera, New York, NY, United States of America

\* paolo.strampelli@gmail.com

## Abstract

Interspecific interactions can be a key driver of habitat use, and must be accounted for in conservation planning. However, spatial partitioning between African carnivores, and how this varies with scale, remains poorly understood. Furthermore, most studies have taken place within small or highly protected areas, rather than in the heterogeneous, mixed-use landscapes characteristic of much of modern Africa. Here, we provide one of the first empirical investigations into population-level competitive interactions among an African large carnivore guild. We collected detection/non-detection data for an eastern African large carnivore guild in Tanzania's Ruaha-Rungwa conservation landscape, over an area of ~45,000 km$^2$. We then applied conditional co-occupancy models to investigate co-occurrence between lion, leopard, and African wild dog, at two biologically meaningful scales. Co-occurrence patterns of cheetah and spotted hyaena could not be modelled. After accounting for habitat and detection effects, we found some evidence of wild dog avoidance of lion at the home range scale, and strong evidence of fine-scale avoidance. We found no evidence of interspecific exclusion of leopard by lion; rather, positive associations were observed at both scales, suggesting shared habitat preferences. We found little evidence of leopard habitat use being affected by wild dog. Our findings also reveal some interspecific effects on species detection, at both scales. In most cases, habitat use was driven more strongly by other habitat effects, such as biotic resources or anthropogenic pressures, than by interspecific pressures, even where evidence of the latter was present. Overall, our results help shed light on interspecific effects within an assemblage that has rarely been examined at this scale. We also demonstrate the effectiveness of sign-based co-occurrence modelling to describe interspecific spatial patterns of sympatric large carnivores across large scales. We conclude by discussing the implications of our findings for large carnivore conservation in modern African systems.

## Introduction

Intraguild competition plays an important role in shaping ecological systems [1–3]. The effects of this form of competition are generally asymmetric, with dominant species having the

**Data Availability Statement:** All data files employed for analysis are publicly available from the Github databases linked to in S1 Appendix

(https://github.com/pstrampelli/Co-occurrence-Input-Files) and S2 Appendix (https://github.com/pstrampelli/Co-occurrence-Model-Rankings).

**Funding:** Scholarship funding for P.S. and funding for some fieldwork costs were provided by the University of Oxford's NERC Environmental Research DTP (https://www.environmental-research.ox.ac.uk/) and by The Queen's College of the University of Oxford (https://www.queens.ox.ac.uk/). Additional fieldwork funding was provided by grants to PS from the National Geographic Society (https://www.nationalgeographic.org/society/), the Columbus Zoological Park Association (https://www.columbuszoo.org/), the Chicago Zoological Society (https://www.czs.org), Panthera (https://panthera.org/), and the Royal Geographical Society (with I.B.G.) (https://www.rgs.org/); we thank them for their generosity. A.D. was funded by a Recanati-Kaplan Fellowship. The funders had no role in study design, data collection and analysis, decision to publish, or preparation of the manuscript.

**Competing interests:** The authors have declared that no competing interests exist.

potential to negatively impact subordinates. This can occur either directly, through intraguild killing and kleptoparasitism [1, 4], or indirectly, by driving spatial, temporal, or dietary shifts to safer but potentially less optimal resources [5, 6]. These competitive pressures can result in a range of multifaceted partitioning mechanisms [2, 7, 8].

Intact African large carnivore guilds feature particularly diverse assemblages of species, all of which are increasingly threatened by rising anthropogenic impacts. In eastern Africa, complete communities include lion (*Panthera leo*, extirpated from approximately 94% of historical range) [9]; leopard (*Panthera pardus*, 48–67%) [10], cheetah (*Acinonyx jubatus*, 91%) [11], spotted hyaena (*Crocuta crocuta*, 24%), striped hyaena (*Hyaena hyaena*, 15%), and African wild dog (*Lycaon pictus*, 93%) [9]. Due to the range of potential interactions, relationships among these sympatric species can play an important role in shaping their habitat use, and thus have important implications for conservation management [4, 6].

Furthermore, large carnivore guilds evolved within large, ecologically heterogeneous ecosystems [2]. However, rising demand for land is increasingly restricting many populations to smaller tracts of habitat [12], which can force increased levels of overlap between competitors, and potentially preclude naturally-evolved partitioning mechanisms. This is compounded by the fact that interspecific interactions are expected to be strongest under conditions of resource limitation [13], and can be altered by human presence [14, 15]. Understanding such interactions in modern, human-impacted systems will therefore be crucial for the successful conservation of large carnivore communities in modern landscapes [16].

A range of mechanisms have been identified that facilitate co-existence between sympatric African carnivores, including partitioning of prey resources [17], space [18], and activity patterns [7]. Some dominant-subordinate dynamics have become increasingly clear; both cheetah and African wild dog (hereafter: wild dog) densities and distributions have been found to vary with lion and spotted hyaena densities [19–21], with such effects suggested as potential threats to cheetah [22, 23]; but see [5, 6, 24] for recent evidence of the contrary and especially wild dog population viability [20, 25]. Competitive pressure from lions on wild dogs is exerted through a combination of kleptoparasitism [26] and direct mortality of both adults and young [27, 28], and there is evidence that wild dogs are excluded from areas heavily used by lions (typically areas with higher prey densities) and that they employ coarser-scale habitat partitioning to coexist with lions [6, 19, 21]. Nevertheless, most studies have been carried out at small spatial scales, rarely investigating population-level avoidance dynamics, and the nature of these displacement mechanisms over large scales therefore remains uncertain. With wild dogs classified as 'Endangered' by the IUCN [29], a more in-depth understanding of these relationships is required to help ensure their persistence in evolving African landscapes.

Research similarly suggests that activity and distribution of leopard can be affected by interspecific competition [5, 30], due to risk of kleptoparasitism, injury, and direct mortality from lions, spotted hyaena, and potentially wild dogs [31]. Although some evidence of interspecific exclusion of leopard by lion exists [32], most studies have found little evidence of avoidance at coarser spatial scales [4, 33, 34]. This suggests finer-scale avoidance dynamics [35], which are likely site-specific variables [5, 32]. Despite leopards sharing approximately 91% of lion range [4], studies examining the relationship between lion and leopard at a population level are limited [34], as are those taking place in human-shaped areas.

Finally, interactions between wild dog or cheetah and leopard have received little research attention, despite the high potential for interspecific competition due to substantial dietary overlap [36, 37]. Although leopards are larger and have been known to kill wild dogs, and the dominance structure between these two species can be fluid [1, 5, 38], the social structure of wild dogs generally places them in a dominant position [36]. However, co-existence mechanics

between these species remain ambiguous [5, 6], and population-level competitive interactions are yet to be studied.

To address these knowledge gaps, we applied two-species conditional occupancy models that account for habitat covariates and imperfect detection [39, 40] to large-scale detection/non-detection sign survey data. We employ these to investigate co-occurrence and co-detection patterns between three threatened sympatric large carnivores (lion, leopard, and wild dog) across Tanzania's ~45,000 km$^2$ Ruaha-Rungwa landscape, to shed more light on population-level interspecific effects between these species (interspecific effects involving spotted hyaena and cheetah could not be determined from our dataset). Two-species occupancy models allow the evaluation of competitive interactions at a landscape scale [41–43], allowing the investigation of population-level mechanisms that cannot be identified through finer-scale methods such as GPS collar studies or dietary analyses [5, 18, 25, 44]. Nevertheless, they have only been used to investigate partitioning between African carnivores on a handful of occasions [34, 45], and never for wild dogs. As intraguild interactions are likely to vary with scale [46], we carry out a set of analyses at two biologically relevant spatial scales. Finally, we investigate how interference competition compares with other factors (biotic resources, anthropogenic impacts) in shaping large carnivore habitat use, in order to contextualise the role of interspecific interactions in shaping large carnivore habitat use.

Overall, our study provides one of the first population-level, multi-scale investigations into large carnivore intraguild competition in a human-impacted, multiple-use African landscape. In doing so, we provide novel insights that can help inform conservation decisions in modern settings. We also show that conditional two-species occupancy models can be applied to sign survey datasets to successfully investigate interspecific effects between large carnivores, albeit with some species-specific limitations.

## Materials and methods

### Study area

The Ruaha-Rungwa conservation complex is a ~45,000 km$^2$ mosaic of protected and non-protected areas, located in south-central Tanzania (Fig 1). The largest protected area (PA) in the system is Ruaha National Park (NP; 20,226 km$^2$), reserved for photographic tourism. The complex also comprises three additional strict PAs which are instead reserved for trophy (sport) hunting tourism (Rungwa, Kizigo and Muhesi Game Reserves (GRs)– 9,175 km$^2$, 5,140 km$^2$, and 2,720 km$^2$, respectively), as well as a number of less-strictly protected areas. These include Lunda-Mkwambi Game Controlled Area (GCA; 1,720 km$^2$) and Rungwa South Open Area (OA; 3,870 km$^2$), where both trophy hunting and additional resource extraction by local communities are permitted, and two community-managed Wildlife Management Areas (WMAs), MBOMIPA and Waga (947 and 344 km$^2$, respectively), where both photographic and hunting tourism are permitted (although neither was taking place at the time of study).

The complex is unfenced, and is surrounded by unprotected village lands to the south and east, and by additional OAs and GCAs to the west and north. Law enforcement is generally higher in the NP and the GRs, due to greater availability and investments of resources, and lower in the GCA, OA, and WMAs [47]. Nevertheless, due to boundary disputes, human settlements and intensive agriculture are present in an area of ~2,100 km$^2$ in south-west Ruaha NP [48]. As a result, the effective boundaries of the NP do not correspond to the gazetted boundaries in this area (Fig 1), which was excluded from our study. Other anthropogenic pressures within the complex include land clearing for settlements and subsistence agriculture, bushmeat poaching, fishing, burning for honey gathering, legal and illegal logging, and illegal mining [48].

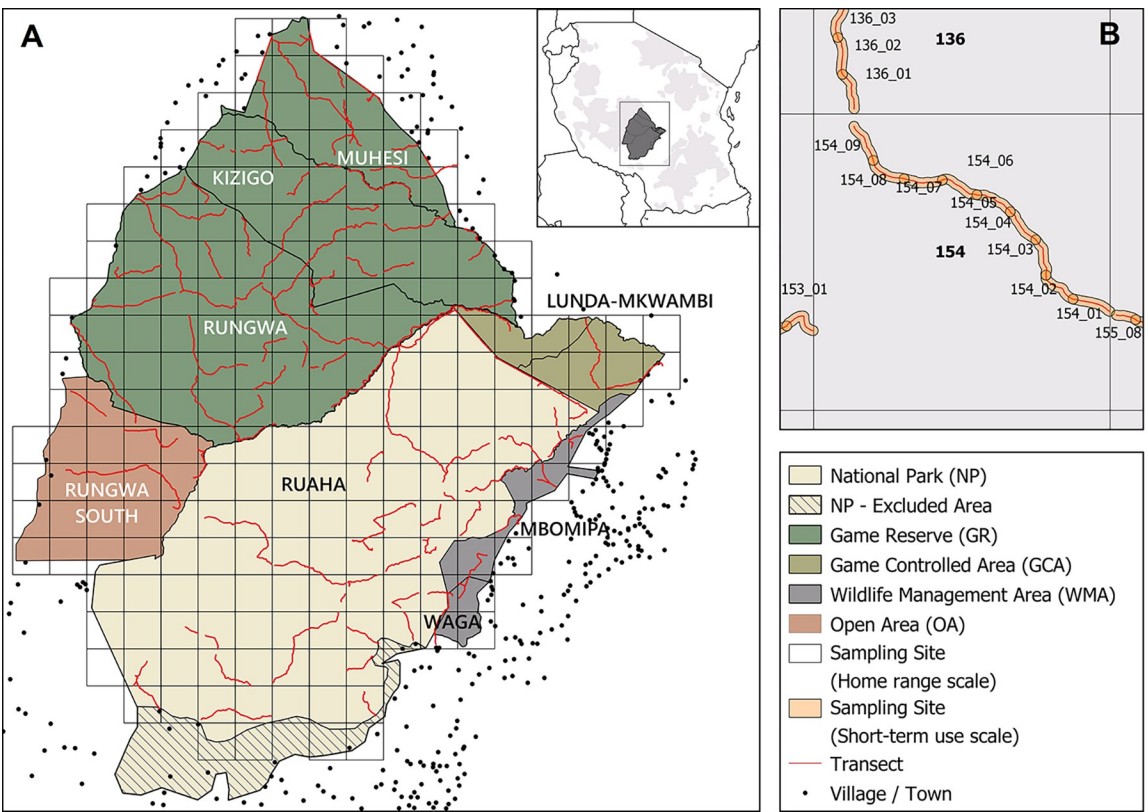

**Fig 1.** The Ruaha-Rungwa conservation landscape (A), within the wider context of Tanzania's protected area system (inset). A: Sampling grid investigating co-occurrence at the home range scale; B: Sampling grid investigating co-occurrence at the short-term use within the home range scale. Sampling sites consist of 225 km$^2$ grid cells at the home range scale, and of an area of ~1 km$^2$ around 2 km survey transect segments at the short-term use scale. Both the gazetted and effective boundaries for Ruaha NP are depicted. Lunda-Mkwambi GCA comprises both Lunda-Mkwambi North and South, while Rungwa South OA comprises both Rungwa South OA and Rungwa Mzombe OA. Only villages in proximity of the PA complex are shown.

Ruaha-Rungwa exhibits a mixed mosaic of *Acacia-Commiphora* open savannah/bushland and central Zambezian and Eastern *Brachystegia*-dominated miombo woodlands [49]. Altitude ranges from 696 m to 2,171 m [50], and the complex is semi-arid, with annual rainfall being highly seasonal (December-April) and varying between 450 and 900 mm across its extent [51]. The landscape harbours a complete eastern African large carnivore guild, including regionally and globally important populations of lion, leopard, cheetah, wild dog, spotted hyaena, and striped hyaena [52, 53].

## Study design

Evaluating competitive interactions at a single scale may lead to misleading inferences [39], and scale-dependency has been hypothesised to explain contrasting patterns of predator suppression in African carnivores [18, 24]. As a result, we investigated interspecific effects at two spatial scales: the first corresponding to home range selection and persistence, and the second to short-term use within the home range. These two scales can also be considered equivalent to investigating long-term (pre-emptive) and short-term (reactive) spatial avoidance [18, 25].

Sampling sites consisted of 225 km$^2$ grid cells at the home range scale. To investigate short-term use within the home range, we divided our home range scale survey transects (see "Data Collection") into 2-km sections, each being considered a short-term use site (Fig 1). Only

short-term use scale sites located within home range scale sites where both species of interest had been detected were included in the short-term use analysis; this ensured that areas with no presence at a coarser scale were not erroneously interpreted as being avoided at the finer scale.

## Data collection

We conducted vehicle-based spoor (track) transect surveys along roads, to collect detection/ non-detection presence/absence data on large carnivores across the landscape. Large carnivores perform extensive movements along road networks [54], and spoor-based surveys have been shown to be an efficient and effective method to collect data on large carnivores over vast landscapes, including for all of our study species [55–57].

Surveys were carried out over two dry seasons, between 7 July and 29 November 2017, and 29 June and 21 November 2018 (we made the assumption that interspecific interactions remained constant across and within seasons). Transects comprising a minimum of 6 km and a maximum of 20 km of roads were surveyed within each home range scale site (differences in sampling effort are accounted for in the modelling process; [40]). The survey team consisted of a driver and two experienced trackers seated on custom-made seats on the bull-bar of the vehicle. The vehicle was driven along a transect at a maximum speed of 10 km/h, and surveying took place between dawn and approximately 10:00 am, before the high-standing sun made detection difficult. If another vehicle travelled on the road during surveying, the transect was ended.

We employed a spatially replicated occupancy sampling approach [47, 55–57]. Each transect was divided into 500 m segments, and we recorded whether sign of each species was detected (1) or non-detected (0) within each segment. Transects were surveyed once, with each site being sampled once, in one of the two survey seasons.

Spoor of five large carnivores (lion, leopard, cheetah, wild dog, and spotted hyaena) were recorded (striped hyaena was not recorded). For all species, data were recorded in the form of detection/non-detection (1 or 0) at the species level within individual 500 m segments. Overall, we were able to survey the majority (85%) of sampling sites in the landscape, thus ensuring representative coverage (Fig 1).

## Co-occurrence modelling

In our modelling framework, sampling occasions were defined as spatially replicated transect segments. Occupancy models that explicitly address the issue of spatial autocorrelation [58] cannot be implemented in two-species analyses. As a result, at the home range scale, segment length was set to account for spatial autocorrelation (Markovian dependence) in detection; this was achieved by testing for this in Program PRESENCE 2.12.37, increasing sampling occasion length until there was no more evidence of spatial autocorrelation [47, 59]. As this was achieved at a length equal to or shorter than 5 km for all species [47], sampling occasions consisted of 5 km transect segments in all analyses at the home range scale. At the short-term use scale, sampling occasions comprised 2–4 continuous 500 m segments, and it was therefore not possible to account for spatial autocorrelation following the method employed at the home range scale. Although a removal sampling design was attempted as an alternative way to account for spatial correlation in detections, the naturally low detectability of large carnivores meant we could not achieve convergence once the necessary data were removed. As a result, it was not possible to account for spatial autocorrelation in detection at the short-term use scale.

We fitted conditional single-season two-species occupancy models [39, 41] in Program PRESENCE 2.12.37 [60] to the sign-based detection/non-detection data. Conditional two-species occupancy models allow the probability of occupancy and detection of one species

(subordinate; species B) to be modelled as a function of the occupancy and detection status of the other (dominant; species A), while controlling for habitat effects [39]. This parametrisation was chosen due to its stability when incorporating covariates [42].

We tested several hypotheses regarding interspecific spatial interactions of three sympatric large carnivores in the study area: lion, leopard, and wild dog. Co-occurrence dynamics between spotted hyaena and cheetah and other large carnivores could not be modelled due to insufficient heterogeneity in detection (spotted hyaena) or detections (cheetah). Lion were considered dominant to wild dog and leopard, and wild dog were considered dominant to leopard [36].

To test our hypotheses on co-occurrence, we estimated the following occupancy parameters at each scale: $\Psi^A$ (occupancy probability of the dominant species), $\Psi^{BA}$ (occupancy probability of the subordinate, when the dominant is present), and $\Psi^{Ba}$ (occupancy probability of the subordinate, when the dominant is absent). We fitted a set of *a priori* models which assumed that the presence of the dominant species influenced the subordinate (i.e. $\Psi^{BA} \neq \Psi^{Ba}$; 'conditional' models), as well as models where the occupancy of the subordinate was independent of presence of the dominant species ($\Psi^{BA} = \Psi^{Ba}$; 'unconditional' models). To determine whether there was evidence of interspecific effects, competing conditional and unconditional models were compared by model ranking based on their Akaike Information Criterion, adjusted for small sample size (AICc). AICc values and model weights acted as a measure of the relative amount of evidence for each model. For each analysis, the conditional model ($\Psi^{BA} \neq \Psi^{Ba}$) receiving the most support would suggest either avoidance ($\Psi^{BA} < \Psi^{Ba}$) or aggregation ($\Psi^{BA} > \Psi^{Ba}$), while the unconditional model ($\Psi^{BA} = \Psi^{Ba}$) receiving the most support would instead suggest that occurrence is independent [39].

To control for habitat effects, and ensure that any identified interspecific effect was not a result of different habitat preferences or detection patterns, we incorporated habitat (site use) and detection covariates from the best ranked model from single-species analyses. These were taken from a study investigating single-species occupancy of large carnivores in the study area [47], and employed the same data and were carried out at the same spatial scales, using the same sampling design. Site use covariates included a suite of biotic, prey, and anthropogenic impacts covariates, while detection covariates related to road quality and use were employed to model differences in detectability during surveying. See S1 Appendix for all input data, including all covariates employed. By fitting models with and without interspecific, site use, and detection covariates, and ranking them against each other, we were able to compare the role of interspecific and habitat effects in shaping occurrence. A general model was employed for detection (p), containing as many parameters as possible (i.e. $pA \neq rA \neq pB \neq rBA \neq rBa$, and the best-ranked detection covariates identified in the single-species model).

To test for interspecific effects on detection, we estimated the following parameters in each analysis: $p^A$ (probability of detecting the dominant species, given the absence of the subordinate), $p^B$ (probability of detecting the subordinate, given the absence of the dominant), $r^A$ (probability of detecting the dominant, given both are present), $r^{BA}$ (probability of detecting the subordinate, given both are present and the dominant is detected), $r^{Ba}$ (probability of detecting the subordinate, given both are present and the dominant is not detected). We compared a set of *a priori models*, based on AICc model rankings, to test whether the detection probability of the subordinate species was either conditional on the occurrence ($p^B \neq r^{BA}$, $p^B \neq r^{Ba}$) or the detection ($r^{Ba} \neq r^{BA}$) of the dominant species, or independent and thus unconditional on the dominant species ($p^B = r^{Ba} = r^{BA}$). In addition, we tested whether the detection of the dominant species was conditional on the detection of the subordinate species ($p^A \neq r^A$) or unconditional ($p^A = r^A$). To control for other effects on detection, we incorporated the best-ranked detection covariates from single-species models [47]. When testing for detection

effects, all models employed the top-ranked model for occupancy, as identified in the previous step [61].

Finally, we also calculated the species interaction factor ('SIF') for occupancy ($\varphi$) and detection ($\delta$) [39] from the top-ranked model for each species pair, at both scales. The SIF represents a likelihood ratio of co-occurrence and co-detection for the two species, estimating how likely the two species are to co-occur compared to what would be expected under a hypothesis of independence [39]. If the SIF = 1, or if the estimate of the SIF ± SE includes 1, the two species occur independently. SIF > 1 suggests that the two species are more likely to co-occur than would be expected under the null hypothesis of independence (aggregation), whereas SIF < 1 indicates spatial avoidance of the dominant species by the subordinate.

Overall, to draw inferences about co-occurrence and co-detection for each species pairing at the different scales of interest, we considered: the estimated parameters ($\Psi^A$, $\Psi^{BA}$, $\Psi^{Ba}$, $r^a$, $p^a$, $p^B$, $r^{Ba}$, $r^{BA}$) of models with strong support ($\Delta$AIC < 2), the relationships between them, model rankings, and SIFs [61–63].

In all analyses, the occupancy estimator ($\Psi$) was interpreted as the probability of site use, rather than that of occupancy, in order to relax the closure assumption and thus minimise potential assumption violations [64].

## Results

We surveyed a total of 2,235 km of transects across eight PAs (Fig 1). A total of 144 sites were sampled at the home range scale (average survey effort per site = 15.5 km), equivalent to ~75% of all sites in the landscape. This resulted in a total of 1484 sites at the short-term use scale.

Overall, we recorded 159 independent detections of lion, 233 of leopard, and 40 of wild dog. We also recorded 11 independent detections of cheetah and 432 of spotted hyaena.

### Large carnivore co-occurrence & co-detection

In all analyses, models which incorporated the effects of habitat (site use) covariates fitted better than models without ($\Sigma$w = 0.99 vs $\Sigma$w < 0.01). In addition, models which included habitat covariates, but not interspecific effects, almost always received significantly stronger support than models which included interspecific effects, but not habitat effects (see S2 Appendix for full model rankings). This indicates that in most cases, and even when significant interspecific effects were identified (see below), habitat use was driven more strongly by other habitat effects than interspecific ones.

### Lion & wild dog interspecific associations

After accounting for habitat and detection effects, we found some evidence of lion avoidance by wild dog at the home range scale; however, these effects were not significant. Although the conditional model ($\Psi^{BA} \neq \Psi^{Ba}$) ranked higher than the unconditional ($\Psi^{BA} = \Psi^{Ba}$), the latter also received strong support ($\Delta$AICc < 2; Table 1). Nevertheless, while not significantly different due to relatively wide standard errors, probability of wild dog use at sites where lion were absent was greater than where lion were present (i.e. $\Psi^{Ba} > \Psi^{BA}$; Table 2, Fig 2). Similarly, the SIF for occurrence suggested non-significant avoidance ($\varphi < 1$; Table 2, Fig 3A).

There was statistically significant evidence for wild dog avoidance of lion at the short-term use scale. Although the unconditional model also received strong support (Table 1), the statistically significant difference between wild dog site use when lion is present or absent ($\Psi^{BA} = 0.16 \pm 0.11$ S.E. and $\Psi^{Ba} = 0.42 \pm 0.14$, respectively; Table 2, Fig 2), indicates that wild dog use of areas within the home range with lion presence was significantly lower. The SIF also suggested a significant avoidance effect ($\varphi = 0.55 \pm 0.31$; Table 2, Fig 3A), strongly

**Table 1. Summary of co-occurrence model rankings used to evaluate the role of interspecific interactions on occurrence of lion, leopard, and wild dog in Ruaha-Rungwa, at two spatial scales.** Models in which site use of the subordinate species depends on the presence (ΨBA) or the absence (ΨBa) of the dominant species (conditional models, denoted as ψBa ≠ ψBA) were compared against models in which site use of the subordinate species is independent of the presence of the dominant species (unconditional models, denoted as ψBa = ψBA). For each analysis, only the best-supported conditional and unconditional models are presented; see S2 Appendix for full model rankings.

| Model | AICc | ΔAICc | W$_i$ | Npars | -2loglike |
|---|---|---|---|---|---|
| **Lion-Wild Dog** | | | | | |
| *Home range scale* | | | | | |
| ΨBA≠ΨBa | 796.12 | 0.00 | 0.56 | 17 | 762.12 |
| ΨBA = ΨBa | 796.65 | 0.53 | 0.44 | 17 | 762.65 |
| *Short-term use scale* | | | | | |
| ΨBA≠ΨBa | 593.39 | 0.00 | 0.45 | 11 | 571.39 |
| ΨBA = ΨBa | 594.68 | 1.29 | 0.23 | 11 | 572.68 |
| **Lion-Leopard** | | | | | |
| *Home range scale* | | | | | |
| ΨBA≠ΨBa | 1187.94 | 0.00 | 0.51 | 16 | 1155.94 |
| ΨBA = ΨBa | 1188.01 | 0.07 | 0.49 | 16 | 1156.01 |
| *Short-term use scale* | | | | | |
| ΨBA≠ΨBa | 3993.13 | 0.00 | 0.55 | 16 | 3961.13 |
| ΨBA = ΨBa | 3993.55 | 0.42 | 0.45 | 16 | 3961.55 |
| **Wild Dog-Leopard** | | | | | |
| *Short-term use scale* | | | | | |
| ΨBA≠ΨBa | 936.79 | 0.00 | 0.4872 | 15 | 906.79 |
| ΨBA = ΨBa | 936.85 | 0.06 | 0.4728 | 15 | 906.85 |

A general model for detection (pA≠rA≠pB≠rBA≠rBa) was employed to model co-occurrence. ΨA = site use of the dominant species; ΨBA = site use of the subordinate species, when the dominant species is present; ΨBa = site use of the subordinate species, when the dominant species is absent. AICc = Akaike Information Criteria, adjusted for small sample size; nPars = number of parameters in the model; W$_i$ = model weight; -2LogLike = twice the negative likelihood of the model.

**Table 2. Site use (ψ), detection probability (p and r), and species interaction factors (SIF–phi and delta) parameter estimates and associated standard errors from the top-ranked model investigating co-occurrence between lion, leopard, and wild dog in Ruaha-Rungwa, at the two spatial scales investigated.**

| | ΨA | ΨBA | ΨBa | pA | rA | pB | rBA | rBa | φ | δ |
|---|---|---|---|---|---|---|---|---|---|---|
| **Lion-Wild Dog** | | | | | | | | | | |
| Home range scale | 0.72 (0.09) | 0.33 (0.11) | 0.58 (0.34) | 0.54 (0.06) | 0.32 (0.07) | 0.10 (0.09) | 0.18 (0.08) | 0.25 (0.09) | 0.77 (0.28) | 0.78 (0.25) |
| Short-term use scale | 0.51 (0.12) | 0.16 (0.11) | 0.42 (0.14) | 0.22 (0.09) | 0.44 (0.15) | 0.30 (0.07) | 0.05 (0.07) | 0.29 (0.19) | 0.55 (0.31) | 0.33 (0.25) |
| **Lion-Leopard** | | | | | | | | | | |
| Home range scale | 0.73 (0.10) | 0.89 (0.05) | 0.86 (0.09) | 0.19 (0.11) | 0.48 (0.05) | 0.46 (0.06 | 0.51 (0.05) | 0.44 (0.04) | 1.02 (0.08) | 1.08 (0.07) |
| Short-term use scale | 0.47 (0.06) | 0.50 (0.06) | 0.45 (0.06) | 0.19 (0.05) | 0.49 (0.03) | 0.42 (0.03) | 0.23 (0.03) | 0.33 (0.04) | 1.06 (0.09) | 0.81 (0.07) |
| **Wild Dog-Leopard** | | | | | | | | | | |
| Short-term use scale | 0.26 (0.07) | 0.38 (0.14) | 0.42 (0.09) | 0.38 (0.08) | 0.36 (0.09) | 0.38 (0.05) | 0.41 (0.11) | 0.21 (0.08) | 0.92 (0.29) | 1.46 (0.34) |

ψA = site use of dominant species; ψBA = site use of subordinate species, when the dominant species is present; ψBa = site use of subordinate species, when the dominant species is absent;; pA = probability of dominant species being detected, when the subordinate species is absent; rA = probability of dominant species being detected, when the subordinate species is present; pB = probability of subordinate species being detected, when the dominant species is not present; rBA = probability of subordinate species being detected, when the dominant species is present and detected; rBa = probability of subordinate species being detected, when the dominant species is present but not detected; φ = SIF for occupancy: ratio of how much more (>1) or less (<1) likely the species are to co-occur at a site compared to what would be expected if the species occurred independently of each other; δ = SIF for detection: ratio of how much more (>1) or less (<1) likely the species are to be detected together in a survey compared to what would be expected if they were detected independently

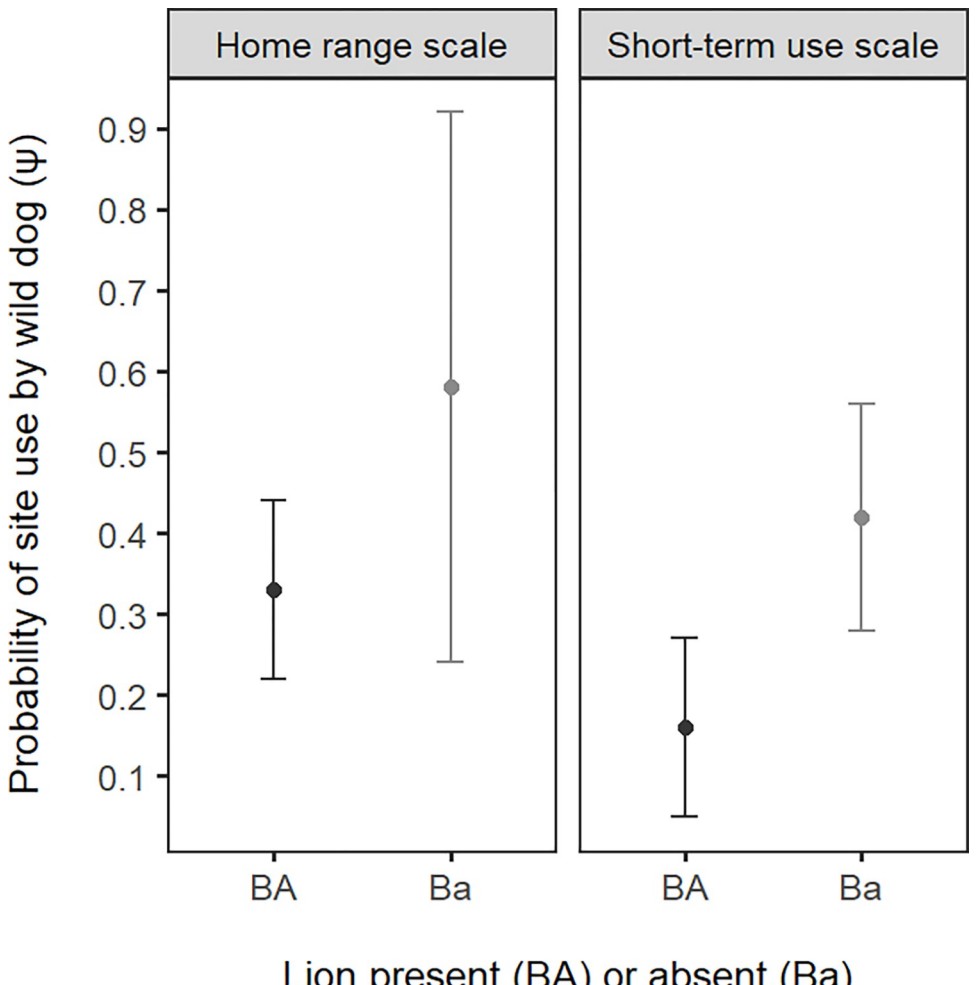

**Fig 2. Mean probability of wild dog site use (with associated standard error) given the presence (BA) or absence (Ba) of lion, from the top-ranked model, at the two spatial scales investigated in Ruaha-Rungwa.**

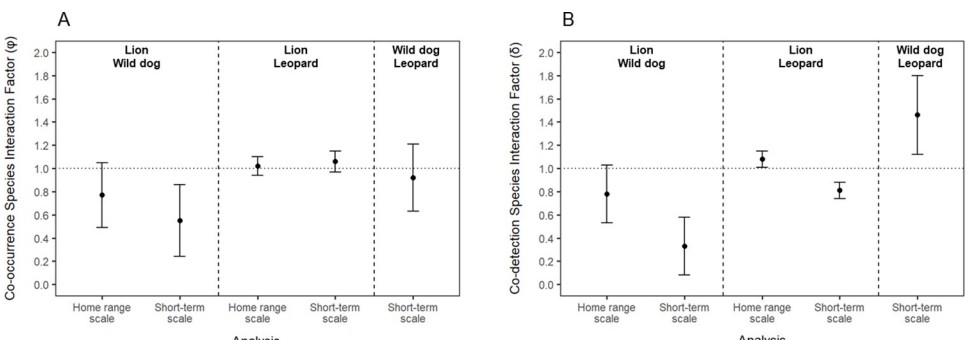

**Fig 3.** Species interaction factors (SIFs; with respective standard errors) representing the level of co-occurrence ($\varphi$; A) and co-detection ($\delta$; B) between lion, leopard, and wild dog in Ruaha-Rungwa, from the top-ranked model, at the two spatial scales investigated. An SIF value exceeding 1 (dotted line) indicates that the species co-occur more often than expected; a value of less than 1 indicates that the species co-occur less often than expected; and a value of 1 indicates that the species are co-occurring randomly.

supporting the hypothesis of wild dog avoidance of lion at this scale. Furthermore, this was the only analysis, of all species pairs and at both spatial scales, in which a model including interspecific effects, but not habitat covariates, received strong support (S2 Appendix). This suggests that avoidance of lion has a particularly strong impact on space use by wild dog within the home range.

With regards to detection, in all analyses at the home range scale the SIF ($\delta$) suggested a negative association between detection of wild dog and lion, although effects were not significant (Table 2, Fig 3B). In addition, although results suggested wild dog detection to be lower when lion are present and detected than when they are present but not detected (rBA < rBa), these effects were also not significant. Model rankings at the home range scale similarly provided only marginal evidence for wild dog detection being different when lions were also present and detected (pB = / = rBA) or present and not detected (pB = / = rBa) compared to when lion were absent, based on the alternative (pB = rBa = rBa) also receiving strong support (Table 3). There was, however, strong support for lion detection being greater when wild dog were absent, based on both parameter estimates (i.e. pA>rA; Table 2) and model rankings (i.e. only models where pA$\neq$rA receiving strong support; Table 3).

Within the home range (short-term use scale), the SIF provided strong evidence for wild dog detection being negatively associated with that of lion ($\delta$ = 0.33 ± 0.25; Table 2, Fig 3B).

**Table 3. Summary of co-occurrence model rankings used to evaluate the role of interspecific interactions on detection of three sympatric large carnivores (lion, leopard, and wild dog) in Ruaha-Rungwa, at two spatial scales (see main text for additional details).** Only models with strong support (i.e. $\Delta$AICc<2) are presented; for full model rankings see S2 Appendix.

| Model | AICc | ΔAICc | $W_i$ | Npars | -2loglike |
|---|---|---|---|---|---|
| **Lion-Wild dog** | | | | | |
| *Home range scale* | | | | | |
| pA≠rA≠pB≠rBA≠rBa | 796.12 | 0.00 | 0.43 | 17 | 762.12 |
| pA≠rA≠pB≠rBA = rBa | 796.77 | 0.65 | 0.31 | 17 | 762.77 |
| pA≠rA≠pB = rBA = rBa | 797.5 | 1.38 | 0.22 | 17 | 763.50 |
| *Short-term use scale* | | | | | |
| pA≠rA≠pB≠rBA≠rBa | 593.39 | 0.00 | 0.30 | 11 | 571.39 |
| pA = rA≠pB≠rBA≠rBa | 593.66 | 0.27 | 0.26 | 11 | 571.66 |
| pA≠rA≠pB≠rBA = rBa | 595.23 | 1.84 | 0.12 | 11 | 573.23 |
| **Lion-Leopard** | | | | | |
| *Home range scale* | | | | | |
| pA≠rA≠pB≠rBA≠rBa | 1187.94 | 0.00 | 0.49 | 16 | 1155.94 |
| pA≠rA≠pB≠rBA = rBa | 1189.57 | 1.63 | 0.22 | 16 | 1157.57 |
| pA≠rA≠pB = rBA = rBa | 1189.72 | 1.78 | 0.20 | 16 | 1157.72 |
| *Short-term use scale* | | | | | |
| pA≠rA≠pB≠rBA≠rBa | 3993.13 | 0.00 | 0.94 | 16 | 3961.13 |
| **Wild Dog-Leopard** | | | | | |
| *Short-term use scale* | | | | | |
| pA≠rA≠pB≠rBA≠rBa | 936.79 | 0.00 | 0.35 | 15 | 906.79 |
| pA = rA≠pB≠rBA≠rBa | 936.82 | 0.03 | 0.34 | 15 | 906.82 |

All detection models included the top-ranked model for site use (see Table 2). pA = probability of dominant species being detected, when the subordinate species is absent; rA = probability of dominant species being detected, when the subordinate species is present' pB = probability of subordinate species being detected, when the dominant species is not present; rBA = probability of subordinate species being detected, when the dominant species is present and detected; rBa = probability of subordinate species being detected, when the dominant species is present but not detected. AICc = Akaike Information Criteria, adjusted for small sample size; nPars = number of parameters in the model; $W_i$ = model weight; -2LogLike = twice the negative likelihood of the model. In all analyses, "=" was used to designate parameters set as equal, while we used "≠" to designate parameters set as different

Moreover, detection of wild dog was greater when lion were absent (pB>rBA/rBa; Table 2), and was greater when lion were not detected than when they were at sites used by lions (rBa>rBA). This was also indicated by model rankings (pB = rBA = rBa not receiving strong support; Table 3).

### Lion & leopard interspecific associations

We found no evidence of interspecific exclusion of leopard by lion at the home range scale, with both model rankings (ΨBA≠ΨBa; Table 1) and parameter estimates (ΨBA>ΨBa, φ>1; Table 2) indicating a weak positive association. Similar association effects were also observed at the short-term use scale (Tables 1 & 2, Fig 3A).

The SIF for detection (δ) suggested significant positive associations in the detection of leopard and lion at the home range scale (δ = 1.08 ± 0.07; Table 2, Fig 3B). There was little evidence of leopard detection being affected by the presence of lion at the home range scale (Tables 2, 3). On the other hand, detection of lion was significantly greater when leopard were present (rA>pA; Table 2), as also supported by model rankings (pA≠rA; Table 3).

Within the home range, on the other hand, results revealed a negative association between detection of lion and leopard (δ = 0.81 ± 0.07; Table 2, Fig 3B), and that detection of leopard was significantly lower when lion were present (pB≠rBA≠rBa being the only model with strong support, Table 3; pB>rBA/rBa, Table 2). Furthermore, leopard detection was greater at sites where lion were present, but not detected, than at sites where lion were present, and also detected (rBa>rBA; Table 2). Finally, detection of lion was greater when leopard were present (pA≠rA, Table 3; rA>pA, Table 2).

### Wild dog & leopard interspecific associations

Due to the widespread detection of leopard and the relatively low number of detections of wild dog, it was not possible to model co-occurrence between these species at the home range scale.

Within the home range, there was little evidence of leopard habitat use being affected by wild dog presence, based on both model rankings (Table 1) and parameter estimates (Table 2, Fig 3A). Regarding co-detection, the SIF (δ) suggested a significant positive association between the two species (δ = 1.46 ± 0.34; Table 2, Fig 3B). The effect of wild dog presence on leopard detection was, on the other hand, inconclusive, as model rankings suggested an effect (pB≠rBA≠rBa; Table 1), but leopard detection when wild dog were not present (pB) was similar to that when wild dog were present and detected (rBA; Table 2). There was also little evidence of wild dog detection differing with leopard occurrence, based on models where pA = rA received strong support (Table 1), and on similar parameter estimates (pA and rA) in the unconditional model (Table 2).

## Discussion

Overall, our findings reveal some multi-scale interspecific avoidance effects, suggesting a particularly strong impact of lion on wild dog space use, and contributing to the increasing body of evidence showing that niche partitioning can occur across multiple spatial dimensions [5].

### Interspecific effects between sympatric carnivores in Ruaha-Rungwa

**Lion & wild dog.** We found some evidence of interspecific exclusion of wild dog by lion at the home range scale. Although these effects were not statistically significant, this may at least in part be due to the relatively low number of wild dog detections (a consequence of the species' wide ranging nature) [65], which likely contributed to the relatively wide standard

errors observed. The same issue was noted when modelling large carnivore co-occurrence in Asia [43]. Given the low number of detections, and the SIF and associated SEs barely overlapping 1 (Fig 3), it appears unlikely that distributions of both species at this scale are independent.

In addition, wild dog detection was lower at sites used by lion, and was negatively associated with that of lion (Table 2, Fig 3). As detection of free-moving animals is influenced by movement rate, use intensity, and local abundance [66, 67], our results suggest that wild dogs are either found at lower abundance in areas used by lions, that they use these areas more rarely, or that they decrease their movements within them [68]. Furthermore, as occupancy modelling does not account for intensity of use or variations in species density [7], it is possible that wild dog disproportionately only avoid areas of high lion density, contributing to the observed lack of significance at this scale. This is supported by single species analysis of the same data [47], and by the fact that lion detection was significantly greater where wild dog were absent, suggesting greater lion densities in these areas. Taken together, our results suggest that lion are likely influencing wild dog spatial patterns, and potentially population dynamics, at a landscape scale.

These findings add to the growing body of literature suggesting spatial displacement of wild dog from areas of high lion density [5, 20, 67, 69, 70]. However, as the models employed cannot directly infer the mechanisms of the underlying ecological interactions [42], the observed effects could, at least in part, be a result of wild dog being unable to persist in areas of high lion use due to increased mortality, rather than solely avoidance [20, 26, 27].

Our modelling also revealed strong evidence of wild dog avoidance of lion within the home range, with wild dog being more than twice as likely to use an area of their home range if this was not also used by lion. Furthermore, the fact that wild dog are less likely to be detected when lions are present or detected suggests that, where they do share home ranges, wild dog exhibit strong fine-scale avoidance of lion (likely through olfactory cues and other fine-scale detection mechanisms) [71]. These finer-scale avoidance effects appear to be particularly strong, with this being the only analysis, for all species pairings and scales, where interspecific effects had a stronger influence than other resources or pressures on habitat use (S2 Appendix).

**Lion & leopard.** Leopard appeared to act independently of lion, with no evidence of interspecific exclusion at either scale. Instead, we observed positive spatial associations between the two species, at both scales. This provides additional evidence that leopard do not spatially segregate from lions [4, 34]. Positive associations of carnivore spatial distributions with those of competitors and even predators have been recorded in other studies, and this might be explained by carnivores occurring at densities that are below a threshold that necessitates species to utilise spatial avoidance as a partitioning mechanism [72, 73]. Co-existence may instead promoted through a combination of dietary partitioning [69], behavioural adaptations (e.g. arboreal caching of prey) [31], and spatiotemporal avoidance at even finer scales than those assessed in our analysis [4, 5]. Indeed, the lower detection of leopard at the short-term use scale when lion were also present suggests avoidance at a very fine spatial scale, likely through a combination of olfactory, acoustic, and visual cues [33, 74]. Our findings thus further indicate that, while leopards do react to lions, they do so at a spatial scale that does not result in broad-scale displacement or prevent leopard from accessing key resources [34]. By responding reactively, and only avoiding preferred habitats or transitioning habitats immediately following lion presence, leopards likely minimise risk of encountering lions while still maintaining access to key resources [5].

On the other hand, the positive co-occurrence patterns observed are likely a consequence of imperfectly modelled shared habitat preferences, with leopard and lion both independently selecting for similar areas of higher prey availability (at the home range scale) and catchability (at the short-term use scale) [75]. The positive associations in detections (at the home range

scale), the fact that lion detection was greater when leopard was present or detected (at both scales), and detection of leopard not being affected by lion presence (at the home range scale), are all similarly plausible consequences of both species selecting for similar areas within their home range, and/or of positively correlated abundances [4, 34].

**Leopard & wild dog.**   Although it was not possible to model wild dog and leopard co-occurrence at the home range selection scale, there was little evidence of leopard habitat use within the home range being impacted by wild dog presence. Although interspecific competition is most prevalent where similarly sized species share similar prey species [1], as is the case for leopard and wild dog [44], the observed lack of effect is likely a result of leopard employing similar adaptations to those employed to coexist with lions [33]. Furthermore, while leopards are mostly nocturnal, wild dog exhibit crepuscular hunting habits [65]. This temporal segregation likely aids co-occurrence, as does the less consistent dominance relationship between these two species, with leopards having been known to kill adult wild dogs [76].

**Relative effect of interspecific competition and other factors in driving habitat use.** Our finding that models with habitat covariates but not interspecific effects almost always received stronger support than those with interspecific effects but not habitat covariates suggests that, with the exception of the effect of lion on wild dog at a finer scale, other biotic and anthropogenic effects play a more important role than interspecific pressures in shaping habitat use of African large carnivores. This is in line with studies elsewhere, which have found that other habitat requirements are generally more important than interspecific competition in structuring carnivore communities, even when evidence of such effects is present [61–63, 77, 78]. Nevertheless, the effect of interspecific competition does appear to vary across studies, suggesting that these forces are complex and context dependent [72, 73].

## Conservation implications

Our results suggest that the successful conservation of wild dog in modern African systems will require sufficient space and habitat heterogeneity to ensure the availability of competition refuges [27]. This is especially the case since wild dog exhibit relatively low tolerances to habitat conversion [29], thus precluding avoidance through the increased use of more anthropogenically-disturbed areas, a strategy employed by some subordinate predators (e.g. leopard with tiger, *Panthera tigris*) [79]. As anthropogenic pressures increase across wild dog range, it will be critical that vulnerable populations are able to minimise detrimental effects of intra-guild interference and predation. Conservation managers must therefore identify and ensure the effective protection of habitats which can serve as competition refuges for wild dog.

Our findings thus highlight the importance of landscape-scale conservation planning for wild dog [80], and the need to include interspecific effects as part of a holistic approach to conservation planning, protected area management, and species recovery programmes [27, 81]. By better understanding species interactions, particularly in light of anthropogenic disturbance, conservation managers will also be able to better predict how system changes may impact carnivore communities in modern systems [82]; such predictions will be particularly relevant in response to global climate change, with rates of range expansion and contraction of species likely to be influenced by co-occurring species [83].

Finally, our findings include some encouraging implications for large carnivore management, in that lion and leopard, and leopard and wild dog, appear to be able to use the same areas without population-level repercussions, even in relatively impacted habitats (although leopards appear to exhibit greater tolerance to human disturbed areas compared to lions in our study area; Strampelli et al., 2022). Knowledge of this should help align management strategies for these species, and streamline conservation interventions [34].

## Methodological considerations & main study limitations

We show that conditional co-occurrence modelling can be successfully employed to study interspecific interactions from detection/non-detection sign survey data in an African setting. Although individual-level interspecific interactions have been extensively studied at finer spatial and temporal scales through radio or GPS-based telemetry [5, 18, 74], direct observation [4, 20, 24], and co-occurrence modelling applied to camera trap data [34, 84], these methodologies often rely on small sample sizes, and are typically not suitable or cost-effective enough to provide the population level insights necessary to inform landscape-scale management [16]. While co-occurrence models have been applied to investigate sympatric carnivore co-occurrence in Asia [43, 85] and the Americas [62, 86], applications to large African carnivores have until now been few [34, 84], and, as far as we are aware, never using sign-based data.

Our study shows that the method holds significant promise to improve understanding of population-level mechanisms in African conservation landscapes, and we encourage similar studies elsewhere, particularly given the existing availability of sign-based detection/non-detection datasets [55, 56, 59, 87, 88]. We especially encourage further investigations into how the effects identified vary across gradients of anthropogenic impacts, and protected area management strategies. Ideally, studies such as ours should be complemented by intensive, finer-scale studies (such as through GPS collar data) [25], as we acknowledge the temporal scale of studies such as ours may be too broad to discern finer-scale interspecific dynamics. Indeed, we believe both methods can provide complementary insights to help inform management. Our findings also further highlight the importance of considering biological scale in habitat use investigations [89, 90], as the true extent of the impact of lion occurrence on wild dog would not have been evident from analyses restricted to a single spatial scale. Finally, our study emphasizes the importance of also accounting for interspecific effects on detection when investigating co-occurrence, as recently highlighted by others [91].

The main limitation of our methodology was the inability to model spotted hyaena and cheetah interspecific interactions. This was due to either low detections (cheetah) or high naïve occupancy and low heterogeneity in use (spotted hyaena). This is a shortfall of this investigation, particularly given the evidence of interactions between spotted hyaena and lion in the study area [84]. We nonetheless believe that the method could be applied to these species in other settings (e.g. areas with higher densities of cheetah, or where spotted hyaena distribution is limited more heavily by anthropogenic or biotic factors). We also encourage the development of conditional co-occurrence models that can account for the lack of independence between spatially replicated sampling occasions, as done for single species models [58]. This would allow future studies to be able to account for this potential source of bias, which we were not able to do at the short-term use scale, and which we acknowledge as being a limitation of our study. While it is difficult to determine how this bias would translate into our findings, we recommend caution when interpreting the short-term use results for this reason.

## Supporting information

**S1 Appendix. Co-occurrence modelling input data.**
(DOCX)

**S2 Appendix. Co-occurrence model rankings.**
(DOCX)

## Acknowledgments

Fieldwork for this research was carried out under permits 2017-210-NA-2017-107 and 2018-367-NA-2017-107, granted by the Tanzania Commission for Science and Technology

(COSTECH) and the Tanzania Wildlife Research Institute (TAWIRI). We would like to thank the Government of Tanzania, TAWIRI, Tanzania National Parks Authority (TANAPA), Tanzania Wildlife Management Authority (TAWA), and Idodi-Pawaga MBOMIPA and Waga WMAs for their support of this research, as well as all the staff of the Ruaha Carnivore Project/ Lion Landscapes. We also thank TANAPA and TAWA Rangers, MBOMIPA Game Scouts, and S. G. Pangamwene and H. S. Dongo for their assistance during fieldwork. We are also grateful to tourism operators who provided logistical assistance when in the field, with a special thanks to Mdonya Old River Lodge and Essential Destinations. We also thank E.J. Milner-Gulland and L. Hunter for the insightful comments on the study.

## Author Contributions

**Conceptualization:** Paolo Strampelli, Philipp Henschel.

**Data curation:** Paolo Strampelli, Charlotte E. Searle.

**Formal analysis:** Paolo Strampelli, Charlotte E. Searle.

**Funding acquisition:** Paolo Strampelli, Amy J. Dickman.

**Investigation:** Paolo Strampelli.

**Methodology:** Paolo Strampelli.

**Supervision:** David W. Macdonald, Amy J. Dickman.

**Writing – original draft:** Paolo Strampelli.

**Writing – review & editing:** Philipp Henschel, David W. Macdonald, Amy J. Dickman.

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
