## [Decision Letter · Decision Letter 0]

26 Oct 2022

PONE-D-22-16672Spatial co-occurrence patterns of sympatric large carnivores in a multi-use African systemPLOS ONE

Dear Dr. Strampelli,

Thank you for submitting your manuscript to PLOS ONE. After careful consideration, we feel that it has merit but does not fully meet PLOS ONE’s publication criteria as it currently stands. Therefore, we invite you to submit a revised version of the manuscript that addresses the points raised during the review process.

We look forward to receiving your revised manuscript.

Kind regards,

Bogdan Cristescu

Academic Editor

PLOS ONE

Journal Requirements:

2. We note that Figure 1 in your submission contain [map/satellite] images which may be copyrighted. All PLOS content is published under the Creative Commons Attribution License (CC BY 4.0), which means that the manuscript, images, and Supporting Information files will be freely available online, and any third party is permitted to access, download, copy, distribute, and use these materials in any way, even commercially, with proper attribution. For these reasons, we cannot publish previously copyrighted maps or satellite images created using proprietary data, such as Google software (Google Maps, Street View, and Earth). For more information, see our copyright guidelines: http://journals.plos.org/plosone/s/licenses-and-copyright.

Additional Editor Comments:

First, apologies for the time it took to provide feedback on this manuscript. Finding reviewers was challenging, but we were finally able to secure two excellent reviews. I concur with their assessment that this work will be an important contribution, once the comments are addressed in detail. Looking forward to the updated manuscript and response to reviewers.

Reviewers' comments:

Reviewer's Responses to Questions

**Comments to the Author**

1. Is the manuscript technically sound, and do the data support the conclusions?

Reviewer #1: Yes

Reviewer #2: Yes

2. Has the statistical analysis been performed appropriately and rigorously? 

Reviewer #1: Yes

Reviewer #2: Yes

3. Have the authors made all data underlying the findings in their manuscript fully available?

Reviewer #1: Yes

Reviewer #2: Yes

4. Is the manuscript presented in an intelligible fashion and written in standard English?

Reviewer #1: Yes

Reviewer #2: Yes

5. Review Comments to the Author

Reviewer #1: Comments to Spatial co-occurrence patterns of sympatric large carnivores in a multi-use African

System (PONE-D-22-16672)

The authors present findings on spatial co-occurrence patterns between three species of the African large carnivore guild in a globally important conservation area. The findings are based on a non-invasive sign survey across a large landscape and build on recent research that identified environmental and anthropogenic drivers of carnivore habitat use in this system. The methodological approach allows multi-species and multi-scale inferences across an area that extends previous efforts. The authors present this method to be valid by reaching similar conclusions to earlier research. This manuscript therefore holds scientific merit, both for our understanding of community dynamics of African large carnivores as well as its potential to be replicated and implemented across other areas with similar needs. My comments are listed below and mainly relate to the structure of the manuscript and some methodological assumptions:

- The manuscript aims to inform large carnivore conservation in modern African systems, yet it lacks some consistency in the introduction of its objectives and the discussion of its findings to be relevant for the large carnivore guild. The results indeed only reflect co-occurrence patterns of the species pairs for which data was available, but one of the key findings – habitat use was driven more strongly by habitat effects than by interspecific pressures – could receive some more attention throughout the manuscript.

- The assumption that leopards are subordinate to wild dogs is questionable and interactions are likely not unidirectional (Curveira-Santos et al. 2021; Palomares & Caro 1999). Is there any information available on mean pack size or personal observations in the system that supports your assumption? Vanak et al. (2013) even showed that all of the African felid species were more likely to move toward wild dogs when in close proximity and therefore concluded that wild dogs were the most subordinate species. I acknowledge that the latter research is based on a modified large carnivore guild in a small reserve, but it could be worth exploring co-occurrence patterns between leopards and wild dogs in two sets of models, one with wild dog as dominant species and one with leopard as dominant species.

Curveira-Santos et al. 2022. Broad aggressive interactions among African carnivores suggest intraguild killing is driven by more than competition. Ecology 103:e03600.

Palomares & Caro 1999. Interspecific Killing among Mammalian Carnivores. The American Naturalist 153:492–508.

Vanak et al. 2013. Moving to stay in place: Behavioral mechanisms for coexistence of African large carnivores. Ecology 94:2619–2631.

L61-82: This section is important to understand current knowledge on interactions between the focal species pairs, but the focal species are only mentioned towards the end of the introduction. This brings some confusion as to why current knowledge on interactions between other species of the large carnivore guild are not presented.

L87-L91: Long sentence, maybe rephrase.

L93-94: Confusing sentence, maybe rephrase.

L120: Typo: The complex i¬s

L139: Closing bracket missing.

L143 (& L233-236): ‘Transect segments’ or ‘transects’? I understand that at the short-term use scale you look at 2 km transects with 500 m transect segments, and at the long-term use scale you look at 6 – 20 km transects with 5 km transect segments? What was the number of sampling occasions at the long-term use scale, especially for sites with only 6 km surveyed? Some clarifications would be in place.

L144-145: Biologically meaningful scale may be species-specific and therefore grid cells of 225 km2 may reflect second order habitat use of lions and leopards but rather third order habitat use of wild dogs. I would rather stick with short and long-term use than second and third order terminology. I acknowledge that the occupancy estimator was interpreted as probability of site use rather than that of occupancy.

L155: Do you have any support that environmental conditions, resource availability and predator populations remained similar across years to assume similar strength of interspecific competition between years?

L167 & L249-250: Was striped hyena not recorded or not detected?

L178-182: Better formulation of your hypotheses would be in place as well as support for wild dog – leopard interaction.

L198-199: A brief summary of the habitat and detection covariates included would be informative.

L233: Format citation

L331: A general discussion preceding the species-pair sections would be in place, in particular with regards to the effect of habitat covariates compared to interspecific pressures and the observation of positive co-occurrence and co-detection trends for some species pairs. Sections L365-367 & L398-402 could be moved and integrated here.

L349: Could this be supported or discussed by the findings from Strampelli et al. 2022. Camera trapping and spatially explicit capture–recapture for the monitoring and conservation management of lions: Insights from a globally important population in Tanzania. Ecological Solutions and Evidence. 3:e1219.

L352 - 364: Another relevant article for your discussion that reaches similar conclusions with different methodology: Goodheart, B., Creel, S., Vinks, M.A. et al. African wild dog movements show contrasting responses to long and short term risk of encountering lions: analysis using dynamic Brownian bridge movement models. Mov Ecol 10, 16 (2022). https://doi.org/10.1186/s40462-022-00316-7

L376-378: Suggestion to rephrase: ‘…they do so at a spatial scale that does not result in broad-scale displacement…’

L403: I feel this section is mainly focused on wild dogs – possibly the species of highest conservation concern – though in the context of your manuscript, I would keep the conservation implications relevant for the large carnivore guild, which can include some species-specific recommendations.

L407- 408: Example available for wild dogs:

Van der Meer et al. 2011. An empirical and experimental test of risk and costs of kleptoparasitism for African wild dogs (Lycaon pictus) inside and outside a protected area, Beh. Eco. 22: 985–992. https://doi.org/10.1093/beheco/arr079

Is there any evidence from your findings that lions displace wild dogs to more anthropogenically-disturbed areas? And would you expect spatial co-occurrence patterns to change with management context?

L421: and leopard and wild dog also.

L431: Though the general patterns are similar regardless of method used.

L446-452: From this and earlier research, both spotted hyena and cheetah appear to be widespread across the landscape. This may provide some indication that their patterns of long-term site use are not influenced by interspecific interactions, which is along the lines of some of your findings and research performed elsewhere.

L452-455: This could indeed introduce a considerable source of bias and it would be valuable to elaborate how spatial autocorrelation in short-term site use may influence your findings. In particular as your spatial autocorrelation test for long-term site use indicated 5 km transect segments, and only 500m segments were used in short-term site use models.

Reviewer #2: This manuscript titled “Spatial co-occurrence patterns of sympatric large carnivores in a multi-use African system” investigates the co-occurrence patterns of a large carnivore guild on two spatial scales. Overall, the manuscript is well presented and provides interesting insights into competition among large carnivore guild members. The methods and occupancy analyses are fairly easy to understand and results are clear. This is an interesting and thorough study that I was pleased to read, as it addresses an important but relatively understudied aspect of carnivore ecology, particularly as it pertains to globally important large carnivore populations and threatened species. The paper is well-written but could benefit from additional clarity in some areas, and the authors could do more to situate these findings in a broader context.

1) The manuscript is technically sound.

2) Statistical analyses have been conducted appropriately

3) Authors has provided findings

4) Manuscript is presented in an intelligible fashion.

6. PLOS authors have the option to publish the peer review history of their article (what does this mean?). If published, this will include your full peer review and any attached files.

Reviewer #1: **Yes: **Stijn Verschueren

Reviewer #2: **Yes: **Willem Daniel Briers-Louw

---

## [Author Response · Author response to Decision Letter 0]

12 Dec 2022

Please see attached response letter.

---

## [Decision Letter · Decision Letter 1]

3 Jan 2023

Spatial co-occurrence patterns of sympatric large carnivores in a multi-use African system

PONE-D-22-16672R1

Dear Dr. Strampelli,

We’re pleased to inform you that your manuscript has been judged scientifically suitable for publication and will be formally accepted for publication once it meets all outstanding technical requirements. Please check references - as Reviewer 2 highlighted some lack of consistency.

Kind regards,

Bogdan Cristescu

Academic Editor

PLOS ONE

Reviewers' comments:

Reviewer's Responses to Questions

**Comments to the Author**

1. If the authors have adequately addressed your comments raised in a previous round of review and you feel that this manuscript is now acceptable for publication, you may indicate that here to bypass the “Comments to the Author” section, enter your conflict of interest statement in the “Confidential to Editor” section, and submit your "Accept" recommendation.

Reviewer #1: All comments have been addressed

Reviewer #2: All comments have been addressed

2. Is the manuscript technically sound, and do the data support the conclusions?

Reviewer #1: Yes

Reviewer #2: Yes

3. Has the statistical analysis been performed appropriately and rigorously? 

Reviewer #1: Yes

Reviewer #2: Yes

4. Have the authors made all data underlying the findings in their manuscript fully available?

Reviewer #1: Yes

Reviewer #2: Yes

5. Is the manuscript presented in an intelligible fashion and written in standard English?

Reviewer #1: Yes

Reviewer #2: Yes

6. Review Comments to the Author

Reviewer #1: The authors addressed the concerns raised during the first round of review by clarifying their ideas and providing sound justifications in the response letter. These explanations enhanced my understanding of the work and the corresponding revisions made in the revised manuscript were clear, albeit in some cases somewhat minor. Overall, these efforts improved the quality of the manuscript and I enjoyed reviewing this well-researched and well-written manuscript. I have no further comments.

Reviewer #2: The authors have addressed all the comments for this manuscript revision. I am happy with their responses and am willing to accept their manuscript for publication. However, the authors should go through the References section again as there are some slight variations in referencing style e.g. Ref 71, 73, 75.

7. PLOS authors have the option to publish the peer review history of their article (what does this mean?). If published, this will include your full peer review and any attached files.

Reviewer #1: **Yes: **Stijn Verschueren

Reviewer #2: **Yes: **Willem Briers-Louw

---

## [Editor Report · Acceptance letter]

10 Jan 2023

PONE-D-22-16672R1 

Spatial co-occurrence patterns of sympatric large carnivores in a multi-use African system 

Dear Dr. Strampelli:

I'm pleased to inform you that your manuscript has been deemed suitable for publication in PLOS ONE. Congratulations! Your manuscript is now with our production department. 

Kind regards, 

on behalf of

Dr. Bogdan Cristescu 

Academic Editor

PLOS ONE